# Social Support Is Linked to Post-Traumatic Growth among Tunisian Postoperative Breast Cancer Women

**DOI:** 10.3390/healthcare10091710

**Published:** 2022-09-07

**Authors:** Feten Fekih-Romdhane, Nihed Riahi, Leila Achouri, Haitham Jahrami, Majda Cheour

**Affiliations:** 1Faculty of Medicine of Tunis, Tunis El Manar University, Tunis 1068, Tunisia; 2Psychiatry Department “Ibn Omrane”, Razi Hospital, Manouba 2010, Tunisia; 3Salah Azaiez Institute, Tunis 1006, Tunisia; 4Department of Psychiatry, Ministry of Health, Manama 410, Bahrain; 5Department of Psychiatry, College of Medicine and Medical Sciences, Arabian Gulf University, Manama 323, Bahrain

**Keywords:** breast cancer, women, post-traumatic growth, spirituality, distress, social support

## Abstract

The struggle with breast cancer (BC) is often associated with positive changes after diagnosis and treatment, that are being referred to as posttraumatic growth (PTG). We aimed to examine PTG and its relationship with psychosocial (i.e., psychological distress, spiritual well-being, social support), sociodemographic and cancer-related variables in Tunisian women operated on for breast cancer. This was a cross-sectional study. Seventy-nine (79) postoperative BC women were administered the Post-Traumatic Growth Inventory, the Spiritual Well-being Scale, the Depression, Anxiety and Stress Scales, and the Multidimensional Scale of Perceived Social Support. The changes most frequently reported by participants were discovering that they were stronger than they thought they were (70.0%), having stronger religious faith (65.0%), and being better able to accept the way things work out (63.8%). Multivariate analysis showed that anxiety and social support remained significantly associated with PTG, while no significant relationship has been found for spiritual well-being. Overall, the present study adds to the existing body of research by identifying factors related to women’ s experience of PTG in a previously unexplored Arab Muslim cultural context, Tunisia. We believe that our findings may help inform strategies aiming at promoting positive psychological changes after experiencing BC, at least in our context.

## 1. Introduction

Breast cancer is the most commonly diagnosed cancer in women in the world (11.7% of total cases) and the leading cause of cancer death [1]. Breast cancer is a life-threatening disease that impairs many life domains [2], and has detrimental effects on women’s physical and psychological health [3].

Despite its traumatic nature, several studies have found that the struggle with this disease was associated with positive changes after diagnosis and treatment, that are being referred to as posttraumatic growth (PTG) [4]. PTG occurs in the aftermath of a highly challenging life event, and consists of changes in several functional aspects of life, including a greater appreciation of life, an enhanced personal strength, improved social relationships, heightened interest in spirituality, an awareness of personal strength, and improved abilities to express emotion and hope for the future [5].

Some previous studies have described PTG among women with oncological illness [6,7]. Indeed, prior research has documented a PTG prevalence of 98% for women with breast cancer diagnosed between 1 and 5.5 years [8]; however, its role in women’s mental health remains unclear.

Several authors have observed that PTG is influenced by many factors, including sociodemographic characteristics, the type of traumatic event [5], perceived social support [9], and spirituality [10]. Particularly, prior research has shown that perceived social support from family, friends, and a private person plays a crucial role in the adjustment for the stresses of breast cancer diagnosis and treatment and in the development of PTG [9]. Also, breast cancer patients who are more spiritual have been found to be more easily able to perceive some positive outcomes from their illness experience, and thus more likely to develop PTG [10]. However, prior research on social support and PTG among cancer patients remains somewhat controversial. While some studies have found that support seeking [11] and received social support [12] were positively associated with PTG; other cancer literature revealed no significant association between these two entities [13]. Similarly, longitudinal research among breast cancer patients in this field is rather scarce and has shown inconsistent results. Furthermore, while there has been a growing interest in PTG in recent years, most of the existing studies on this topic have originated from Western countries. However, it has been shown that PTG is a culturally dependent concept; with cultural background playing a key role in shaping the trauma experience and the subsequent behavior adopted by the individual to deal with it [14]. The non-universality of the PTG concept calls for international research from under-researched non-Western countries. To our knowledge, very few or no studies investigating PTG among cancer patients have been performed in Arab countries from the MENA region.

We aimed, in the present study to examine PTG in a sample of postoperative breast cancer women. Our second aim was to investigate the relationship between PTG (as the dependent variable) and a range of psychosocial (i.e., psychological distress, spiritual well-being, social support), sociodemographic and cancer-related variables (as independent outcomes) in the same sample. In light of the previous literature (e.g., [10,15,16]), we hypothesized that women with lower psychological distress, more spiritual well-being, social support and more severe clinical presentation of breast cancer would exhibit more PTG.

## 2. Materials and Methods

### 2.1. Sample and Procedure

This was a cross-sectional survey that took place over a period of 6 months, from 1 July 2019 to 31 December 2019, at the outpatient surgery department of (anonymized). Inclusion criteria were: (1) being a breast cancer woman outpatient, aged between 18 and 65 years old, (2) having a histologically confirmed breast cancer diagnosis (according to Tumor-Node-Metastasis [TNM] classification), (3) being informed of the breast cancer diagnosis by their treating physicians, and (4) having undergone surgical treatment for their cancer at least one month before the survey.

Non-inclusion criteria were the following: (1) being hospitalized during the time of the survey, and (2) having been previously diagnosed with a mental health disorder (according to the Diagnostic and Statistical Manual of Mental Disorders, fifth edition).

A trained researcher who was a medical doctor specializing in surgical oncology [NR] introduced the survey and provided consistent instructions on how to complete the questionnaire. Twenty female patients refused to participate in our survey. Ninety patients were interviewed, among them eleven were excluded because of a desire to discontinue participation in the study or for incomplete answers. Finally, 79 women with breast cancer having received surgical treatment were included in this study.

As for the ethical considerations, we clearly explained to the participants the objectives of the survey in order to obtain participants’ free and informed consent. Participants’ data confidentiality has been ensured. The study protocol has been approved by the Institutional Review Board of the Faculty of Medicine of Tunis.

### 2.2. Measures

All potential participants were administered a self-report questionnaire that contained two sections.

A first section included socio-demographic (residency, socioeconomic status, educational level, occupation, marital status, living arrangement and stressful life events during the last 12 months) and clinical information (personal somatic history, breast cancer stage, tumor size, cancer treatment received, nature of the surgery, cancer recurrence, and self-perceived pain, fatigue and cognitive problems intensity). Respondents were asked to indicate their occupational status as follows: (1) employee refers to being employed under a contract of employment, (2) worker was defined as working without a steady income, (3) unemployed, and (4) retired. We measured the self-rated economic level by using the question: “How would you rate your current economic level?” The responses were based on a three-point Likert item as follows: low, average, or high. Missing clinical data were collected from the patients’ medical files.

A second section of the questionnaire included four research instruments: the Post-traumatic Growth Inventory (PTGI), the Spiritual Well-being Scale (SWBS), the Depression, Anxiety and Stress Scales (DASS-21), and the Multidimensional Scale of Perceived Social Support (MSPSS).

The Arabic PTGI [17], initially developed by Tedeschi and Calhoun [5], includes 21 items scored from 0 (“I did not experience this change as a result of my crisis”) to 5 (“I experienced this change to a very great degree as a result of my crisis”), passing through 1 (a very small degree), 2 (a small degree), 3 (a moderate degree), 4 (a great degree). The PTGI measures five specific domains of posttraumatic growth: relating to others, new possibilities, personal strength, spiritual change, and appreciation of life. We used Schroever’s and Teo’s method [18] in order to examine detailed changes experienced by participants. Thus, responses to each PTGI item were dichotomized. For each item, an initial score of “not at all experienced” to “experienced a very small degree” is transformed into a score of 0 (we considered that the change linked to the item is not experienced), while an item with an initial score of “experienced to a small degree” to “experienced to a very great degree” is transformed into a score of 1 (we considered that the change linked to the item has been experienced). The sum of the 21 items gives an overall PTG score. The 21 items are grouped into five dimensions including relating to others, new possibilities, personal strength, spiritual change, and appreciation of life.

The SWBS [19] comprises 20 items: ten items measure religious well-being, and ten other items measure existential wellbeing. The SWBS gives two subscores: (1) a cumulative score for religious well-being items, and (2) a cumulative score for existential well-being items. The two subscale scores range from 10 to 60. Higher scores indicate greater religious or existential well-being, respectively. The scale was used in its Arabic version [20].

The DASS-21 [21] is a 21-item self-administered tool composed of three scales (depression, anxiety and stress), with seven items per scale. The subject indicates to what extent each item corresponds to it, on a scale of Likert ranging from 0 (no symptoms) to 3 (symptom present intensively). An overall score is obtained by multiplying the scores by 2. The following cut-off scores have been used to define mild/moderate/severe/extremely severe scores for each DASS scale: Depression: normal (0–4), mild (5–6), moderate (7–10), severe (11–13), extremely severe (14+); Anxiety: normal (0–3), mild (4–5), moderate (6–7), severe (8–9), extremely severe (10+); Stress: normal (0–7), mild (8–9), moderate (10–12), severe (13–16), extremely severe (17+). The Arabic version of the DASS-21 was used in this study [22].

The MSPSS [23] provides assessment of three sources of perceived social support: family, friends, and significant other. Each item is rated on a 7-point Likert rating scale (1 = Very strongly Disagree, 7 = Very strongly agree), with higher scores indicating higher perceived social support. A cut-off of 65 or less was used to define low levels of perceived social support. An Arabic validated version of the MSPSS [24] was used in this study.

### 2.3. Statistical Analysis

Data were analyzed using SPSS, ver. 24 (SPSS Inc., Chicago, IL, USA). Distribution of continuous variables (PTG, SWBS, DASS-21, MSPSS scores, and age) was compared to normal distribution via the Shapiro–Wilks test. Our variables were normally distributed; so parametric tests were used. Therefore, to respond to our second objective, we carried out *t*-tests and one-way ANOVA to determine whether there were any statistically significant differences between PTGI scores (as a continuous dependent variable) and socio-demographic and clinical data (as categorical independent variables). In addition, bivariate correlations between PTGI scores and other continuous variables were investigated using Pearson product-moment correlation coefficients. Finally, multiple regression analyses were carried out to explore the variables related to the PTG outcome among women after a surgical treatment of breast cancer. Variables associated with PTGI scores in the univariate analysis (*p* < 0.05) and those considered as important based on literature review were included in the multivariate analysis. In all statistical tests, the significance threshold was set at 0.05.

## 3. Results

### 3.1. Sociodemographic and Clinical Characteristics of Participants

Women were aged between 29 and 65 years, with a mean age of 52.7 ± 9.8. As seen in Table 1, 71.3% were unemployed and 73.8% were married. Respondents had their surgery a mean duration of 76.4 ± 56.4 months (11 to 252) previously, with a mean duration since cancer diagnosis disclosure of 80.0 ± 55.9 months (12 to 253). Mean scores and standard deviations of study variables are illustrated in Table 2.

### 3.2. Descriptive Statistics for PTGI Items

Regarding post-traumatic growth, the changes most frequently reported by participants were discovering that they were stronger than they thought they were (70.0%), having stronger religious faith (65.0%), and being better able to accept the way things work out (63.8%) (Table 3).

### 3.3. The Associations between PTGI and Study Variables

Univariate analyses (Table 1) have shown that fatigue was significantly and positively associated with PTG (*p* = 0.033). No significant association was found between PTG and other clinical variables. Pearson correlations (Table 4) found a significant positive association between PTGI total scores and anxiety (r = 0.301, *p* = 0.007) and social support (r = 0.321, *p* = 0.004). On the other hand, participants’ age was negatively correlated with post-traumatic growth (r = −0.264, *p* = 0.019).

We then performed stepwise multiple linear regression analysis to determine the factors associated with PTGI symptoms (Table 5). The significant variables were anxiety (t = 2.06, *p* = 0.043) and social support (t = 2.776, *p* = 0.007), and both exhibited medium effect size changes in posttraumatic growth (standardized coefficients of 0.298 and 0.296, respectively).

## 4. Discussion

To the best of our knowledge, this study was the first to explore PTG and examine its related psychosocial, clinical and sociodemographic factors using a multivariate analysis in Tunisian women operated on for breast cancer. We expected that lower psychological distress, more spiritual well-being, more social support and more severe clinical presentation of breast cancer would be associated with more PTG in our sample. Our hypothesis was only partially supported, showing that that social support was the strongest factor related to PTG in our postoperative breast cancer women. No association remained significant between spiritual well-being and PTG in the multiple linear regression analysis. This indicates that perceived social support may be linked to positive outcomes among breast cancer women after surgery, and that spiritual resources seem to have no significant value in their PTG levels. These results are in line with those of previous studies that have examined the role of perceived social support in PTG among clinical populations who have survived life-threatening illness [25,26], as well as among nonclinical populations after traumatic events [27,28]. According to Tedeschi and Calhoun [5], social support (particularly that received from a partner, family or friends) plays a major role in promoting PTG through an activation of cognitive processes, especially with regard to finding meaning in the aftermath of a life crisis. In accordance with our findings, Schroevers et al. [12] and Scrignaro et al. [29] found a positive link between social support and PTG. However, Danhauer et al. [30] noted that, over the first 24 months following diagnosis, higher levels of social support were found in breast cancer women reporting both the highest and the lowest PTG. The latter inconsistent finding may be explained by the fact that a high social support may prevent a negative impact on breast cancer survivors’ mental health that generates growth [30].

Linear regression analysis also found a significant positive relationship between PTG and anxiety, while no significant relationship was objectified between depression, stress and growth. Prior meta-analyses on the relationship between psychological distress and PTG have yielded controversial results. While Zoellner and Maercker [31] reported no significant link between these two variables, Helgeson et al. [32] concluded that individuals who experience greater growth had lower levels of depression. A recent systematic and critical review among people with cancer (46% focused on breast cancer, and 76% used the PTGI) found that growth was inversely or not related to anxiety, to depression, and to distress; with PTSD being the only psychiatric condition that exhibited a direct link with PTG [33]. Other research in breast cancer samples has produced inconclusive results concerning the effects of growth on distress. Some investigations found that growth may minimize the mental health consequences of the disease, leading to less distress and depression [6,34]; whereas others did not objectify any association between growth and distress [35]. Previous longitudinal research found that PTG would contribute to a positive view of life leading to lower distress [36]. Inversely, other authors claimed distress would be positively associated with PTG, because individuals would have recourse to growth to minimize the negative effects of the disease [7]. In summary, the relationship between distress and growth among breast cancer survivors is likely complex and bidirectional in nature. Future research is needed to further examine the relationship between distress and growth in breast cancer women.

Otherwise, we found no significant relationship between spiritual well-being and PTG in the multivariate analysis. Contrary to our finding, several previous studies have found a positive association between spirituality and growth [37,38,39]. Review data regarding cancer populations that focused on this relationship, although it is limited, have reported a direct positive relationship between PTG and spiritual well-being [33]. Prior research in breast cancer patients documented that higher spirituality predicts higher growth [10]. However, in line with our results, some studies do not support this positive association [40,41], and others even claimed that spirituality could be detrimental [42]. Indeed, spirituality may be beneficial when it involves strategies such as praying for others’ well-being, seeking forgiveness and being forgiven, redefining cancer as beneficial, and solving problems in collaboration with God [43]. On the other hand, spirituality may generate negative effects to the practitioner when it comes to feeling dissatisfaction with God, delegating problem solving to God, questioning God’s existence, and redefining cancer as divine punishment [44]. While the current findings suggest that breast cancer may have enabled women patients with cancer internal growth and transformation regardless of their spirituality, the variability in available literature data and the cross-sectional nature of this study highlights a need for further investigation. Coping strategies in times of adversity may vary across culture [45]. Spirituality has been shown to be a way of life in Western cultural contexts; with differences between Western and Middle Eastern countries having been highlighted in health and spiritualty [46]. This suggests a need for additional studies exploring whether cancer can lead to PTG through spirituality in other MENA countries, to confirm or refute our findings. Finally, we did not observe statistically significant association of Illness-related variables with PTG among our participants. In line with our findings, meta-analytic results found no relationship between cancer site, cancer recurrence, severity of illness, stage, and type of treatment received [32].

A few limitations should be discussed. First, the cross-sectional design of the study does not allow for any conclusion about causation. Future longitudinal studies regarding growth after breast cancer and using larger samples are required to ascertain the relationship between growth, distress, social support and illness-related variables. Second, the presence of fatigue, pain and cognitive complaints was assessed using a single-item question and without use of measurement instruments. Similarly, we assessed participants perceived self-rated economic status subjectively, using a three-point single-item scale. Further studies should address these limitations. Third, we used the PTGI to assess patients’ PTG levels. While this is a valid and widely-used measure, it contains items about spiritual change (e.g., “I have a better understanding of spiritual matters”) as well as social interaction and personal relationships (e.g., “I have a greater sense of closeness with others”); which could have influenced the associations between PTG, spirituality and social support. Finally, the assessment of psychological distress, social support and growth was done at different moments after the diagnosis of cancer; however, adjustment to cancer may differ and vary according to evolution [47].

## 5. Conclusions

Overall, the present study adds to the existing body of research by identifying factors related to women’ s experience of PTG in a previously unexplored Arab Muslim cultural context, Tunisia. We believe that our findings may help inform strategies aimed at promoting positive psychological changes after experiencing BC, at least in our context. Although future longitudinal research is needed, our results add support to the major role of social networks and social support in generating positive psychological experiences and likely in buffering negative outcomes after breast cancer. Except for a few small studies, clinical research in this regard is still lacking. Further studies on this research area are required, but generally, the summary of findings highlight the crucial role of social support to better understand the mechanisms that facilitate adjustment to breast cancer, and the importance of interventions that enhance supportive relationships to promote growth among survivors.

## Figures and Tables

**Table 1 healthcare-10-01710-t001:** Sociodemographic and clinical characteristics of participants (*n* = 79).

	N (%)	PTGI * Scores	
		Mean (SD)	*p* Value
Residency			0.860
Rural	21 (26.3%)	54.5 (24.7)	
Urban	58 (72.5%)	53.6 (19.3)	
Socioeconomic level			0.745
Low	11 (13.8%)	58.2 (21.4)	
Medium	66 (82.5%)	53.0 (21.0)	
High	2 (2.5%)	56.0 (1.4)	
Educational level			0.377
Primary	56 (70.2%)	50.1 (22.0)	
Secondary	20 (25.0%)	58.5 (16.4)	
University	3 (3.8%)	57.3 (4.5)	
Occupation			0.242
Unemployed	57 (71.3%)	51.0 (20.2)	
Worker	7 (8.8%)	65.1 (20.4)	
Employee	9 (11.3%)	58.4 (21.7)	
Retired	6 (7.5%)	60.5 (21.9)	
Marital status			0.603
Single	12 (15.0%)	55.0 (24.0)	
Married	59 (73.8%)	54.7 (20.8)	
Divorced	4 (5.0%)	40.2 (11.4)	
Widow	4 (5.0%)	51.7 (16.4)	
Living arrangement			0.688
Alone	3 (3.8%)	57.3 (15.5)	
With spouse	54 (67.5%)	54.9 (20.6)	
With parents	22 (27.5%)	50.6 (21.9)	
Stressful life events during the last 12 months			0.413
No	60 (75.0%)	54.9 (21.3)	
Yes	19 (23.8%)	50.4 (19.0)	
Personal somatic history (other than cancer)			0.283
No	42 (52.6%)	56.3 (19.9)	
Yes	37 (46.3%)	51.2 (21.8)	
Clinical characteristics			
Stage of cancer			0.377
0	6 (7.5%)	42.2 (12.0)	
I	9 (11.3%)	54.7 (20.5)	
II	45 (56.3%)	56.5 (20.6)	
III	19 (23.8%)	50.7 (22.7)	
Tumor size			0.831
≤20 mm	26 (32.5%)	52.7 (17.7)	
>20 mm and ≤50 mm	44 (55.0%)	55.0 (23.0)	
>50 mm	9 (11.3%)	51.1 (18.0)	
Chemotherapy			0.093
No	11 (13.9%)	44.1 (19.2)	
Yes	68 (86.1%)	55.4 (20.6)	
Radiotherapy			0.141
No	5 (6.3%)	40.6 (18.4)	
Yes	74 (93.7%)	54.7 (20.7)	
Hormone therapy			0.879
No	11 (13.9%)	54.7 (21.7)	
Yes	68 (86.1%)	53.7 (20.7)	
Nature of the surgery			0.729
Conservative	18 (22.5%)	55.3 (23.0)	
Radical mastectomy	61 (76.3%)	53.4 (20.2)	
Cancer recurrence			0.583
No	76 (95.0%)	54.1 (21.0)	
Yes	3 (3.8%)	47.3 (10.8)	
Pain			0.482
Nonexistent	65 (81.3%)	52.6 (20.6)	
Mild to moderate	13 (16.3%)	58.9 (21.5)	
Severe	1 (1.3%)	68.0)	
Fatigue			**0.033**
Nonexistent	50 (62.5%)	50.1 (20.8)	
Mild to moderate	29 (36.3%)	60.3 (19.2)	
Severe	-	-	
Cognitive complaints			0.452
Nonexistent	66 (82.5%)	52.6 (21.3)	
Mild to moderate	12 (15.0%)	59.3 (17.4)	
Severe	1 (1.3%)	(69.0)	

* SD: standard deviation; PTGI: Post-Traumatic Growth Inventory.

**Table 2 healthcare-10-01710-t002:** Mean scores and standard deviations of study variables.

	Min	Max	Mean	SD
**Scores**				
**Social support (MSPSS *)**	12	84	62.9	17.2
**Depression (DASS *)**	0	21	9.5	7.4
**Anxiety (DASS)**	0	21	4.7	5.8
**Stress (DASS)**	0	39	11.0	8.2
**Religious well-being (SWBS *)**	36	50	40.7	2.3
**Existential well-being (SWBS)**	30	48	39.5	4.2
**PTGI * total score**	0	99	53.8	20.7
**Relating to others (PTGI)**	0	34	15.7	8.6
**New possibilities (PTGI)**	0	25	8.4	6.4
**Personal strength (PTGI)**	0	20	15.3	4.9
**Spiritual change (PTGI)**	0	10	6.1	3.2
**Appreciation of life (PTGI)**	0	15	8.4	4.5

* SD: standard deviation; PTGI: Post-Traumatic Growth Inventory; MSPSS: Multidimensional Scale of Perceived Social Support; DASS: Depression Anxiety Stress Scale; SWBS: Spiritual Well-Being Scale. Bold values: significant at *p* < 0.05.

**Table 3 healthcare-10-01710-t003:** Percentage of women who answered “I experienced this change as a result of my crisis to a small degree” to “to a very great degree” to each of the PTGI items.

PTGI Items	Number (%)
1. I changed my priorities about what is important in life.	20 (25%)
2. I have a greater appreciation for the value of my own life.	38 (47.5%)
3. I have developed new interests.	17 (21.3%)
4. I have a greater feeling of self-reliance.	45 (56.3%)
5. I have a better understanding of spiritual matters.	22 (27.5%)
6. I more clearly see that I can count on people in times of trouble.	21 (26.3%)
7. I established a new path for my life.	6 (7.5%)
8. I have a greater sense of closeness with others.	23 (28.8%)
9. I am more willing to express my emotions.	16 (20.0%)
10. I know that I can handle difficulties.	36 (45.0%)
11. I can do better things with my life.	43 (53.8%)
12. I am better able to accept the way things work out.	51 (63.8%)
13. I can better appreciate each day.	39 (48.8%)
14. New opportunities are available which wouldn’t have been otherwise.	6 (7.5%)
15. I have more compassion for others.	35 (43.8%)
16. I put more effort into my relationships.	16 (20.0%)
17. I am more likely to try to change things that need changing.	24 (30.0%)
18. I have stronger religious faith.	52 (65.0%)
19. I discovered that I’m stronger than I thought I was.	56 (70.0%)
20. I learned a great deal about how wonderful people are.	27 (33.8%)
21. I better accept needing others.	18 (22.5%)

**Table 4 healthcare-10-01710-t004:** Pearson correlation between PTGI total scores and other continuous variables (*n* = 79).

	r	*p* Value
**Depression**	0.071	0.537
**Anxiety**	0.301	**0.007**
**Stress**	0.124	0.275
**Social support**	0.321	**0.004**
**Religious well-being**	0.071	0.533
**Existental well-being**	0.194	0.087
**Age**	−0.264	**0.019**

Bold values: significant at *p* < 0.05.

**Table 5 healthcare-10-01710-t005:** Multiple linear regression analyses correlating PTG to study variables in our sample (*n* = 79).

Variables	Standardized Bêta Coefficient	*t*	*p* Value	95% CI
Age	−0.199	−1.844	0.069	(−0.88; 0.03)
Fatigue	0.086	0.737	0.463	(−6.23; 13.55)
Depression	−0.212	−1.000	0.321	(−1.78; 0.59)
Anxiety	0.298	2.064	**0.043**	(0.04; 2.10)
Stress	0.186	0.907	0.368	(−0.56; 1.51)
Religious well-being	0.025	0.167	0.868	(−1.03; 1.21)
Existential well-being	0.192	1.470	0.146	(−0.27; 1.78)
Social support	0.296	2.776	**0.007**	(0.10; 0.60)

Bold values: significant at *p* < 0.05.

## Data Availability

The data presented in this study are available on request from the corresponding author.

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
