# Peer review of "Social Support Is Linked to Post-Traumatic Growth among Tunisian Postoperative Breast Cancer Women"

_healthcare, 2022, doi:10.3390/healthcare10091710_

Round 1
Reviewer 1 Report (Previous Reviewer 2)
I am happy with the amendments and changes provided. Well done on making the changes required, it is an interesting paper.
Author Response
Please see the attachment

Reviewer 2 Report (New Reviewer)
The authors presented a cross-sectional study. This is a type of design where researcher assess the association between the exposure and the outcome at the same time. The authors should clearly discuss about 1-exposure(s) 2- Outcome(s) 3-Reasecrch question and (4)Hypothesis.
Line 23-24:
Using cross sectional design, you cannot establish trend over long-term.
Line 25-26:
Is it really you suggest? Please rephrase according to your research question, hypothesis and your result.
Line 65-66:
1- Are these two different objectives?
Objective needs clarity. The author is interested in assessing the relationship between PTG, psychological distress, spiritual welcome, and social support with ----- what?
2- What is (are) your exposure? what is (are) your outcome? Mention clearly.
if there is no exposure / outcome then what are you assessing by doing a cross sectional study? What is your hypothesis? or what is your research question?
3- Why have you done t test? ANOVA? is there any treatment you are looking between the groups OR within the groups?
4- What is your hypothesis?
Line 67:
is this the objective? does not seem so. therefore, delete it from here.
Line 72-73:
how about post operative BC women?
Line 83:
Here, your target population is different than what you have stated above in "inclusion criteria". This should be similar.
Additionally, write this statement. Because these women are those who received some type of treatment for BC (at least this is what I have understood). and not those women with BC (this is what you have written).
Line 94:
Delete this because BC is not the socio-demographic variable.
Line 101-102
Low, average, high economic level- is not it subjective? you should mention objective measure e.g. mention income level or any other objective measurement to measure current economic level.
but you are including BC diagnosed and treated patients (post-operated BC women, you have stated this in your objective). Is not it? So why are you again diagnosing BC?
Line 146:
1-needs clarity.
2. enlist continuous variables
3. what is compared with what?
4. why comparison is needed?
5- is this your objective? If yes, then clearly state. If not, remove this from here because normal distribution of the study variables is needed to decide which test(s) is needed to assess the risk estimates.
Line 147:
1- why you need test test / ANOVA?
2- are you comparing different treatments? if so, this is not clear from your objective.
Line 150:
1- Enlist here
2- What is your exposure and outcome?
3- regression analysis is not used to isolate predictors, instead, it is used to assess the risk of having an exposure on the outcome.
Therefore, author needs to clearly state what risk factors are associated with the outcome in your study.
Line 155:
It seems you excluded these cases. See your exclusion criteria and revise.
Table 1:
Why have you calculated the P-value of these variables? is this your objective? or you want to observe that these variables are not significantly different at baseline? or in two / three groups? If yes, then what are your treatment groups? OR what are you testing at pre-and-post level?
The section 3.2 line 170 and after it recommend to thoroughly revision base on my comment.
Table 4:
Standard beta coefficient should interpreted and discuss in results.
Line 199:
why have you done regression? what association are you looking into? clearly mention.
what are you predicting using cross-sectional data? clearly write.
what is your hypothesis?
Line 259-261:
It seems for me a part of introduction and not the conclusion of your study.
Author Response
Please see the attachment

This manuscript is a resubmission of an earlier submission. The following is a list of the peer review reports and author responses from that submission.
Round 1
Reviewer 1 Report
Thank you for inviting me to review ‘Social support is linked to post-traumatic growth among Tunisian postoperative breast 2 cancer women’. This work assessed the association of PTG with other psychosocial changes in n=79 women with breast cancer cross-sectionally. My comments are below:
Introduction:
1.The first sentence cites work form 2008, please include a more up to date citation/reference and prevalence estimate.
- PTG is defined as being made up of several aspects including “a greater appreciation of life, an enhanced personal strength, improved social relationships, heightened interest in spirituality, an awareness of personal strength, and improved abilities to express emotion and hope for the future”. Later authors state that PTG is influenced by “sociodemographic characteristics, the type of traumatic event [5], perceived social support 48 [9], and spirituality [10]”. Please elaborate more on how PTG can be influenced by the same characteristics that make-up PTG, particularly social support and spirituality.
Materials and Methods:
- Why was 65 years old the age cut-off?
- 4. Please clarify the eligibility criteria around #4 “having had a 60 surgical treatment of the cancer since at least one month”
- Please explain why these measures were interviewer administered and not self-administered?
Results:
- Please expand on the definitions of the occupation categories.
Discussion:
- The main finding that social support was the only statistically significant item in the model after covariate adjustment is not very surprising given that social relationships are a component of PTG, especially since this is cross-sectional data. The authors mention “positive changes” but this measure is retrospective and the social support measure assesses current social support. It needs to be more clearly stated that the PTI items could be influencing the social support. As well as the other measures (spirituality and depression anxiety).
Reviewer 2 Report
Thank you for inviting me to review this manuscript. It is an interesting survey on posttraumatic growth among women with diagnosis of breast cancer. The authors should be congratulated on conducting this experiment, however to improve the manuscript I would suggest the following.
INTRODUCTION
It would be helpful to see more recent rates and prevalence information on breast cancer (authors are mentioning data in 2008).
To make this section stronger, I would advise the authors to highlight more the gap in current literature which brought them to run this study.
METHODS
Please add information on ethics approval.
Line 74, the authors should specify the training and background of the interviewer.
Line 74, minor error “to sections”. I presume it should be two.
Line 80 “TNM classification”, please make sure to specify the meaning of the abbreviation.
Questionnaires used are generally well presented, however it would be beneficial to explain the aim of the PTGI (what does it measure?) by being more concise in the presentation of the answers.
Authors should clarify if timing of data collection from the survey and stage of the recovery have an association.
RESULTS
It is not entirely clear how the classification for socioeconomic status was performed, authors should add more information on this. Also, I would advise authors to change the word “good” for socioeconomic status to high.
I would suggest authors to split table 1 in two different tables, there are quite a lot of information in there. It would be easier for the reader to have separate tables.
To make this section clearer, I would advise authors to create subsections with subheadings according to the different outcomes presented (demographics, clinical characteristics, mental health).
Line 174 “contributed significantly”, please make sure to report p value in the text too.
Was an unadjusted model performed before adjusting?
DISCUSSION
Authors should make sure to present findings in paragraphs.
Line 239, please make sure to be consistent with references. Authors should remove (Paredes & Pereira, 2018).
Authors should discuss more the role of spirituality. Line 241 it could be detrimental, how? How timing is important? How this apply to your study?
Authors should discuss also the potential role of culture within their sample. In which kind of samples were collected data in previous studies? Was the sample population? Authors acknowledge the presence of cultural influences and differences.
CONCLUSION
“Like a violent thunderclap in a serene sky”, I would avoid this kind of language in scientific writing.
Round 2
Reviewer 2 Report
Well done on getting ready the paper with the changes, I would advise for future review to avoid providing answers only as "done". It would be much easier for the reviewer to see changes directly in the answer file.
